# In Search of Sleep Biomarkers of Alzheimer’s Disease: K-Complexes Do Not Discriminate between Patients with Mild Cognitive Impairment and Healthy Controls

**DOI:** 10.3390/brainsci7050051

**Published:** 2017-04-29

**Authors:** Flaminia Reda, Maurizio Gorgoni, Giulia Lauri, Ilaria Truglia, Susanna Cordone, Serena Scarpelli, Anastasia Mangiaruga, Aurora D’Atri, Michele Ferrara, Giordano Lacidogna, Camillo Marra, Paolo Maria Rossini, Luigi De Gennaro

**Affiliations:** 1Department of Psychology, Sapienza University of Rome, Rome 00185, Italy; flaminia.reda@uniroma1.it (F.R.); maurizio.gorgoni@uniroma1.it (M.G.); giulia.lauri@uniroma1.it (G.L.); ilaria.truglia@uniroma1.it (I.T.); susanna.cordone@uniroma1.it (S.C.); serena.scarpelli@uniroma1.it (S.S.); anastasia.mangiaruga@uniroma1.it (A.M.); aurora.datri@uniroma1.it (A.D.); 2Department of Biotechnological and Applied Clinical Science, University of L’Aquila, L’Aquila 67100, Italy; michele.ferrara@univaq.it; 3Institute of Neurology, Catholic University of The Sacred Heart, Rome 00168, Italy; g.lacidogna@libero.it (G.L.); camillo.marra@policlinicogemelli.it (C.M.); paolomaria.rossini@policlinicogemelli.it (P.M.R.); 4Istituto di Ricovero e Cura a Carattere Scientifico, San Raffaele Pisana, Rome 00163, Italy

**Keywords:** amnesic mild cognitive impairment, Alzheimer’s disease, sleep, K-complexes, EEG, cognitive decline

## Abstract

The K-complex (KC) is one of the hallmarks of Non-Rapid Eye Movement (NREM) sleep. Recent observations point to a drastic decrease of spontaneous KCs in Alzheimer’s disease (AD). However, no study has investigated when, in the development of AD, this phenomenon starts. The assessment of KC density in mild cognitive impairment (MCI), a clinical condition considered a possible transitional stage between normal cognitive function and probable AD, is still lacking. The aim of the present study was to compare KC density in AD/MCI patients and healthy controls (HCs), also assessing the relationship between KC density and cognitive decline. Twenty amnesic MCI patients underwent a polysomnographic recording of a nocturnal sleep. Their data were compared to those of previously recorded 20 HCs and 20 AD patients. KCs during stage 2 NREM sleep were visually identified and KC densities of the three groups were compared. AD patients showed a significant KC density decrease compared with MCI patients and HCs, while no differences were observed between MCI patients and HCs. KC density was positively correlated with Mini-Mental State Examination (MMSE) scores. Our results point to the existence of an alteration of KC density only in a full-blown phase of AD, which was not observable in the early stage of the pathology (MCI), but linked with cognitive deterioration.

## 1. Introduction

It is well known that Alzheimer’s disease (AD), the most frequent age-related neurodegenerative disorder [1,2], is characterized by several sleep changes [3] associated with cognitive decline: neural degeneration can induce these sleep alterations, which in turn may provoke a worsening of the cognitive decline, e.g., by the impairment of sleep-dependent memory processes [4]. A relevant evidence of the relationship between sleep and the neurodegenerative process has been recently provided in an animal model by the study of Xie et al. [5], that points to the central role of sleep in neurotoxic waste products clearance. In particular, sleep is associated with a 60% increase in the interstitial space between neural cells, resulting in an enhanced β-amyloid (a protein that plays a key role in AD pathogenesis) clearance rate. Moreover, Mander et al. [6] have shown an association between the β-amyloid burden within the medial prefrontal cortex, an impairment of Slow Wave Activity (SWA) generation, and an impairment of overnight memory consolidation in a sample of healthy older adults.

Given this bidirectional influence between sleep disturbances and AD symptoms, the identification of sleep changes in mild cognitive impairment (MCI) [7,8,9], a clinical condition which is considered a possible transitional stage between normal cognitive function and probable AD [2], could be a relevant line of research.

An alteration often observed in AD patients is a topographically diffuse slow-wave activity characterizing both sleeping and waking electroencephalogram (EEG) [4,10] that makes difficult the distinction between sleeping and waking states in these patients. It seems peculiar that the K-complex (KC), one of the principal hallmarks of Non-Rapid Eye Movement (NREM) sleep, often considered as a “prototype” of reactive sleep slow waves [11], has received scarce attention in the field of sleep research in AD. KC represents the EEG graphoelement with the highest amplitude during normal sleep. It can occur spontaneously but it can also be evoked by different kinds of stimuli [12,13], and is characterized by a negative slow sharp wave followed by a positive slow wave [13,14] with a frontal predominance [13,15,16] and a duration of ≥0.5 s. Converging evidence in animals and humans points to a cortical origin of the KCs [17,18,19,20], while the thalamus seems to have a role in mediating the cortically generated KCs [18,21].

The functional meaning of KCs is not yet clear. Several authors consider them as arousal-related phenomena [22,23], based on the observation that a KC can be elicited by different sensorial stimuli, and given its similarity with an evoked potential [16,24]. Others suggest KCs have a sleep protective role [13,19,25,26,27]: in fact, KC density increases prior to transition to slow-wave sleep (SWS) compared to transition to Rapid Eye Movements (REM) sleep [28] and decreases across consecutive sleep cycles [29]. KCs are also more likely to occur during a recovery night after a night of fragmented sleep [26]. Several authors have highlighted the two-sided nature of KCs, representing a low-level information processing on the one hand and a sleep-preserving mechanism on the other [21,30,31,32,33]. Several studies suggest that normal aging is associated with a decrease of spontaneous KCs [34,35] and an incidence and amplitude decrease of evoked KCs [36,37,38]. Regarding AD patients, early observations have found a reduction of spontaneous KCs [39,40,41]. However, most of these studies were characterized by a small sample size. Moreover, only Montplaisir and co-workers [41] have measured KC density expressed as the number of KCs corrected by NREM stage 2 sleep duration. The choice of this index does not seem trivial due to the observation of differences in polysomnographic features between AD and a healthy elderly control (HC). Furthermore, a decrease in number and amplitude of evoked KCs, and a lower probability of eliciting a KC related to higher dementia severity was reported in AD patients [42]. Finally, we have recently studied frontal KCs in 20 AD patients and 20 healthy age-matched controls (HCs) describing a drastic decrease of KC density during stage 2 NREM in AD patients compared to HCs. AD patients showed a more than 40% reduction in KC density. This drop allowed a correct classification of 80% [43].

All these findings point to an alteration of spontaneous and evoked KCs in AD, but none provides information about when, in the development of AD, the drastic decrease in the rate of KCs starts. In this respect, fast spindles, which represent another phasic event of NREM sleep, showed a significant reduction not only in AD but also in MCI patients as compared to healthy controls [44]. Moreover, sleep spindles density was positively correlated with MMSE scores [44].

Therefore, we decided to also measure KC density in MCI patients. Although these patients have several sleep alterations [45], to the best of our knowledge KCs have never been evaluated in MCI patients. To this aim, we compared the KC density of MCI patients to the measures collected in our AD and HC samples [43].

## 2. Materials and Methods

### 2.1. Subjects

In the present study, 20 amnesic MCI patients (8 males and 12 females; mean age: 72.20) were recruited and compared to our previous sample composed of 20 AD patients and 20 HCs (see [41] for more information). The MCI group consisted of two subgroups, corresponding to two clinical subtypes: 10 single-domain (SD) and 10 multi-domain (MD) MCI patients. The SD type is characterized by a selective memory impairment, whereas in the MD type the deficits are present in at least one other cognitive domain in addition to memory [46]. Demographic and clinical characteristics of the sample are reported in Table 1. Patients were selected among the elderly persons referred to the Neuropsychology Unit of the Gemelli Catholic University Hospital of Rome. HCs were recruited from clubs for retired people.

All subjects gave their written informed consent. The study was approved by the local Institutional Ethics Committee and was conducted in accordance with the Declaration of Helsinki.

### 2.2. Inclusion and Exclusion Criteria

All participants underwent a cognitive screening by means of the Mini Mental State Examination (MMSE) [47]. Moreover, the State Trait Anxiety Index (STAI-Y1 and STAI-Y2) [48] and the Hamilton Depression Rating Scale (HDRS) [49] were administered in order to exclude major psychiatric illness.

Neuropsychological investigation for MCI, as well as for AD patients, included a structured clinical evaluation, brain neuroimaging (Magnetic Resonance Imaging-MRI or Computed tomography-CT), and a neuropsychological test battery for the assessment of specific cognitive functions such as memory, attention, executive function, visuo-construction abilities and language. In particular, memory assessment included Rey’s Auditory Verbal Learning Test (RAVLT) [50], involving immediate recall (RAVLTir), delayed recall (RAVLTdr) and delayed recognition (RAVLTrec), delayed recall of the Rey figures [51], delayed recall of a three-word list [52] and delayed recall of a story [53,54]. The functional status was assessed by the Activities of Daily Living/Instrumental Activities of Daily Living (ADL/IADL) questionnaire [55]. AD patients were included according to the National Institute on Aging-Alzheimer’s Association workgroups [56] and Diagnostic and Statistical Manual of Mental Disorders, fourth edition, (DSM-IV) criteria. 

People with amnesic MCI were enrolled according to guidelines and clinical standards [57,58,59,60].

Common exclusion criteria for all participants were: the presence of neurological, psychiatric or vascular disorders, obesity, history of alcoholism or drug abuse. The final enrolment in the study was based on the evaluation of regular sleep–wake cycle and on the absence of self-rated sleep disorders. The presence of other sleep disorders was objectively evaluated by nocturnal sleep recordings. In case of sleep disorder and/or respiratory diseases and obstructive sleep apnoea syndrome (OSAS), subjects were excluded by subsequent analyses. Although we selected subjects who had less than five events with oxygen saturation <90% per hour of sleep, this however did not exclude the possibility of hypopnoeas due to the lack of a standard evaluation of sleep apneas. Sleep quality and diurnal sleepiness of all participants were assessed by the Italian version of the Pittsburg Sleep Quality Index (PSQI) [61], the Epworth Sleepiness Scale (ESS) [62] and the Karolinska Sleepiness Scale (KSS) [63].

### 2.3. Study Design

Participants underwent a complete polysomnographic (PSG) recording of a nocturnal sleep. A Micromed system plus digital polygraph was used for the PSG recording. EEG signals were acquired with a sampling frequency of 256 Hz and band-pass filtered at 0.53–40 Hz. The 19 unipolar EEG derivations of the international 10–20 system (C3, C4, Cz, F1, F2, F3, F4, F7, F8, Fz, O1, O2, P3, P4, Pz, T3, T4, T5, T6) were recorded from scalp electrodes with average mastoid references (A1 and A2), using Ag/AgCl electrodes. Electro-oculogram (EOG) was recorded from electrodes placed about 1 cm from the medial and lateral canthi of the dominant eye. Electrocardiogram (ECG) and submental electromyogram (EMG) were also recorded. Finally, a pulse oximeter was placed on the right index finger with the aim of excluding sleep respiratory disorders. Impedance was kept below 5 KOhm. 

### 2.4. Data Analysis

#### 2.4.1. Demographics and Clinical Characteristics

Age, years of education and clinical characteristics (MMSE, HDRS, STAI Y-1, STAI Y-2 and PSQI scores) of AD, MCI and HC groups were compared by means of one-way analyses of variance (ANOVAs), and post hoc comparisons were carried out by means of unpaired two-tailed *t*-tests. Alpha level was always set at 0.05.

#### 2.4.2. Sleep Measures

Sleep stages of the baseline night were scored visually in 20-s epochs, according to standard criteria [64], excluding ocular and muscle artefacts. The following were considered as dependent variables: (a) stage 1 latency; (b) stage 2 latency; (c) total sleep time (TST), defined as the sum of time spent in stage 1, stage 2, SWS and REM; (d) percentage of each sleep stage (time spent in a sleep stage/TST × 100); (e) wakefulness after sleep onset (WASO), in minutes; (f) number of awakenings; (g) number of arousals; (h) total bed time (TBT); and (i) sleep efficiency index (SEI = TST/TBT × 100). An awakening was scored whenever an EEG/EMG activation occurred lasting more than 10 s. Arousals have been scored whenever an EMG activation affected the EEG recording for periods shorter than 10 s. 

The polysomnographic EEG measures were submitted to one-way ANOVAs comparing AD, MCI, and HC, and *post hoc* comparisons were carried out by means of unpaired two-tailed *t*-tests.

#### 2.4.3. KC Detection and Analysis

Spontaneous KCs in the three groups were visually identified by a blind scorer during NREM stage 2 sleep on Fz cortical derivation (in line with [38]), just after each nightly recording. To score a KC, the same criteria of the previous study [43] were applied: a non-stationary event with (a) a marked and well-delineated initial negative sharp wave, immediately followed by a positive component; (b) a maximum amplitude at frontocentral derivations; (c) a minimum duration of 0.5 s and a maximum duration of 3 s. In accordance with Crowley et al. [38], no amplitude criterion was applied in the present study, due to the previously observed amplitude decrease in older subjects [37]. If multiple KCs appeared in sequence, only the first one was considered [65]. KC density was calculated as the number of KCs divided by NREM stage 2 sleep minutes. Group difference in KC density was assessed by means of one-way ANOVA comparing AD, MCI, and HC groups, and post hoc comparisons were carried out by means of unpaired two-tailed *t*-tests. Preliminary analyses have also considered gender as a between factor, without any significant main effect or interaction involving this factor. For this reason, it was collapsed in the subsequent analyses. 

Finally, in order to assess the relationship between KC and cognitive impairment, Pearson’s correlation coefficient was computed between KC density and MMSE scores.

## 3. Results

### 3.1. Demographic and Clinical Characteristics

Results of the one-way ANOVAs and relative post hoc *t*-tests performed on demographic and clinical characteristics of AD, MCI and HC groups are reported in Table 1. A significant between-groups difference was observed for MMSE scores: post hoc *t*-tests showed that MMSE scores were significantly higher in HCs compared with AD and MCI patients; moreover, significantly higher MMSE scores have been found in MCI compared with AD patients. Another significant difference was found in the RAVLT immediate (*t* = 4.11, *p* = 0.0002) and delayed (*t* = 3.1, *p* = 0.004) recall between AD and MCI patients. AD had worse performances than MCI patients in both tests. Data on these tests were available only for the clinical samples. No significant difference was observed for age, education, STAI Y1, STAI Y-2, HDRS or PSQI. 

### 3.2. Sleep Measures

Table 2 reports the results of the one-way ANOVAs and post hoc *t*-tests performed on PSG measures. A significant difference was found for SWS, and post hoc *t*-tests show a higher percentage of SWS in HCs compared to both AD and MCI patients. 

### 3.3. KC Density

Figure 1 depicts KC density in AD, MCI and HC groups. The result of the one-way ANOVA shows a significant difference in KC density between groups (*F*_2,57_ = 8.07, *p* = 0.0008), and the post hoc *t*-tests index shows a lower KC density in AD patients compared with HCs (*t* = 3.7, *p* = 0.0007) and MCI patients (*t* = 3.42, *p* = 0.0015). To ascertain whether the absence of statistical differences between HCs and MCI patients was determined by the clinical heterogeneity of the MCI group, a control analysis was performed, considering the two MCI subgroups and comparing 10 single-domain (SD) and 10 multi-domain (MD) MCI patients. The group difference in KC density was assessed by means of one-way ANOVA comparing AD, MD-MCI, SD-MCI and HC groups, and the result shows a significant difference in KC density between groups (*F*_3,56_ = 5.290564, *p* = 0.0028). Fisher’s Least Significant Difference (LSD) post hoc tests showed a significantly lower KC density in AD patients (X = 0.47; SE = 0.06) compared to MD-MCI patients (X = 0.81; SE = 0.03; *p* = 0.007), SD-MCI patients (X = 0.79; SE = 0.1; *p* = 0.01) and HCs (X = 0.82; SE = 0.07; *p* = 0.0008). No significant differences between the MD- and SD-MCI groups (*p* = 0.89) nor between MD-MCI patients and HCs (*p* = 0.93) or SD-MCI patients and HCs (*p* = 0.81) were found.

### 3.4. Correlation between KC Density and Cognitive Impairment

A significant positive correlation (*r* = 0.38, *p* = 0.003) has been observed between MMSE scores and KC density (Figure 2).

## 4. Discussion

To the best of our knowledge, the present study is the first to describe AD-related changes in KCs, also taking into account MCI patients. According to our results, the KC reduction does not seem associated with the MCI condition. On the other hand, the amount of SWS discriminates HCs from both clinical groups. While no differences have been observed between MCI and HC groups, KC density was positively related with MMSE scores, which represent a measure of the degree of cognitive decline.

Our previous study [43] showed a large decrease of frontal spontaneous KC density in AD patients compared with HCs. This fall of KCs does not appear in the MCI group, not even after dividing it into two subgroups (SD and MD). In line with converging evidence, SD and MD may represent two phases along a continuum between normal aging and AD [66,67], so a progressive reduction of KCs between these phases would be expected. On the contrary, the present data suggest that the fall of KCs seems to have a tardy onset, appearing only in a full-blown phase of the disease. A possible explanation for the absence of KC density alteration in MCI patients is that the brain damage responsible for the KC density reduction emerges only in diagnosed AD. Alternatively, such brain alterations could be already present in MCI patients, but they are not severe enough to induce a significant KC density decrease. 

At the same time, these data co-exist with two findings that discriminate HCs from both clinical groups: the amount of SWS and of fast sleep spindles, the other EEG hallmark of stage-2 NREM sleep. With respect to the former, the current study and the study by Gorgoni et al. [44] substantially show a higher percentage of SWS in HCs compared to MCI and AD patients, without any difference between the clinical groups. On the other hand, parietal fast sleep spindles density decreases in both MCI and AD patients compared to HCs, while it does not differ between MCI and AD groups [44]. Hence, it seems that both KC and spindle density, as well as the amount of SWS, decrease in AD patients compared to controls, but the lessening occurs at different stages of disease (earlier for sleep spindles and SWS). The time lag between these phenomena may reflect different pathological mechanisms underlying alterations in AD of KCs on the one hand, and of spindles and SWS on the other. This temporal dissociation could potentially shed light on the AD-MCI neural substrates’ degenerative process. 

The spindle generation mainly involves the interaction between GABAergic inhibitory neurons of the thalamic reticular nucleus and thalamocortical networks [68,69,70,71,72]. At the same time, the pivotal role of the thalamocortical neurons for the generation of delta activity, which mainly characterizes SWS, has been widely demonstrated [70,73,74].

On the other hand, converging evidence in animals and humans points to a cortical origin and propagation of KCs [17,18,19,20], while the thalamus seems to have only a secondary role in mediating the cortically generated KCs [18,21]. Concurrently, several findings point to a sequential deterioration process in different brain areas in AD [67,75,76]: grey matter atrophy, characterizing AD and MCI, mainly implicates bilateral medial temporal lobe (MTL) (parahippocampus gyrus, amygdala, hippocampus, entorhinal cortex, uncus), posterious cingulate cortex, precuneus and thalamus, while more extensive cortical regions (including the frontal, temporal, parietal and insular areas) and subcortical areas seem to be altered only in AD [77]. Moreover, according to the evidence of the time course of neuropathological alterations of neurofibrillary tangles in AD [78] and MCI [79], the neuronal changes seem to start from the locus coeruleus and the dorsal raphe nucleus [80,81], and to spread at first through the MTL and afterwards, and only at the moment of the full diagnosis of AD, to the frontal, parietal and temporal cortical areas and more subcortical regions [77].

The temporal dissociation between the fall of KCs, and sleep spindles and SWS, then, could be explained by the fact that, in the progression of AD, the cortical areas implied in the KC generation mechanism may undergo a deterioration in a later phase of the degenerative process compared to the thalamic and cortico-thalamic pathways implicated in sleep spindles and delta activity production. As a consequence, in MCI patients, the frontal cortex is probably still intact enough to be able to produce KCs.

Even if this explanation remains speculative, the present findings seem to be coherent with the possible, extensively reported in literature, time course of the disease, which characterizes the deterioration of different brain areas in the progression of AD. 

Finally, the absence of significant differences between the three groups in the WASO and SEI does not confirm some previous findings [3,4]. This unexpected result should likely be interpreted as a consequence of our strict exclusion criteria relative to the presence of sleep disorders. 

An important limitation of the present study is linked to the issue of the diagnosis of MCI. According to the National Institute on Aging and Alzheimer’s Association’s workgroup [82], impairment in episodic memory is most commonly seen in MCI patients who subsequently progress to a diagnosis of AD. However, the cognitive evaluation on which the diagnosis is based is not able to determine if the primary cause of the symptoms are degenerative or not (for example, they could be vascular, depressive, traumatic, due to medical comorbidities or mixed disease), and thus it is not able to determine if the MCI condition is due to AD or to other underlying causes. Moreover, our study does not report data about the main well-known biomarkers of AD, such as β-amyloid and tau neurofibrillary tangles, that could have provided fundamental hints about MCI etiology. Probably, different neural substrates could determine different outcomes in terms of sleep changes and, in particular, relative to the drop in KCs. For this reason, we are carrying out a follow-up study in order to be able to select only the MCI subjects who convert to AD. Removing from the analysis the MCI subjects whose symptoms are due to other pathological conditions may potentially shed light on the likelihood of the relationship between the fall of KCs and, specifically, the pathology of AD.

As a final point we have to specify that, although we excluded from the sample subjects with a diagnosis of obstructive sleep apnoea syndrome (OSAS) or with more than five events with oxygen saturation <90% per hour of sleep, the absence of a standard Obstructive Sleep Apnea (OSA) evaluation should be considered a further limitation, as participants could have had hypopneas unrevealed by the oximetry measurement.

## 5. Conclusions

Our findings suggest that no KC density alteration characterizes the sleep of MCI patients. Moreover, KC density is positively associated with MMSE scores, suggesting a role of KC alterations in the process of cognitive decline. Albeit the present findings seem to exclude the possibility of considering KC density as an early biomarker of AD, it could provide some interesting hints about the possible connection between different stages of the AD neurodegenerative process and specific sleep alterations. Future studies should be addressed to systematically investigate the neural basis and the functional role of the AD-related KC density decrease. It would be necessary to compare the EEG with neuroimaging data to better assess the possible relationship between KC alterations and the neural deterioration level of different brain regions in several stages of the disease. 

Given the bidirectional influence between cognitive deterioration and sleep alterations in AD, a better knowledge about the timeline of sleep modifications in these patients could be useful from a clinical point of view, (1) for endorsing methods for an early identification of the MCI condition, and (2) in order to better understand how the improvement of specific sleep features in different stages of the disease could help to reduce the AD-related cognitive deterioration process.

## Figures and Tables

**Figure 1 brainsci-07-00051-f001:**
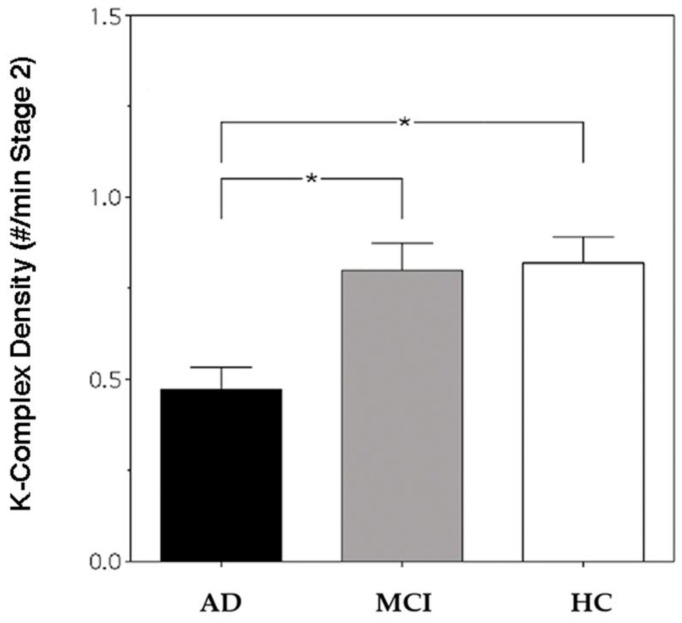
K-Complex (KC) density of Alzheimer’s disease (AD) patients (black bar), mild cognitive impairment (MCI) patients (grey bar) and healthy controls (HCs) (white bar) at Fz cortical derivation. Error bars represent the standard errors. Asterisks (*) indicate statistically significant differences (*p* ≤ 0.05) between groups after post hoc unpaired *t*-tests.

**Figure 2 brainsci-07-00051-f002:**
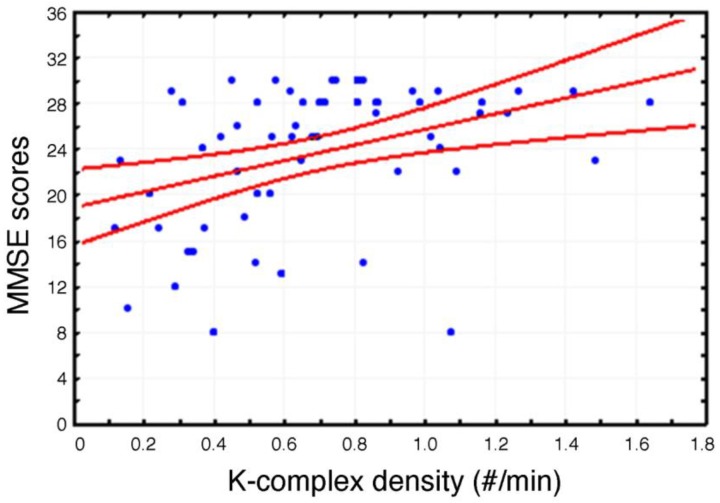
Scatterplot of the individual correlations between Mini Mental State Evaluation (MMSE) scores and K-Complex (KC) density at Fz cortical derivation (*p* ≤ 0.05). Pearson’s r and relative *p* value are reported.

**Table 1 brainsci-07-00051-t001:** Mean and standard errors (SE) of demographic (age, education) and clinical (MMSE, HDRS, STAI Y-1, STAI Y-2, PSQI, RAVLTir, RAVLTdr) characteristics of Alzheimer’s disease (AD) patients, amnesic Mild Cognitive Impairment (MCI) patients and healthy controls (HCs). The results of the one-way Analyses of Variance (ANOVAs) (*F* and *p* values) were also reported, with post hoc unpaired *t*-test (*p* values) when ANOVAs were significant (*p* ≤ 0.05). Significant differences between groups are indicated in bold. The scores at the Ray’s Auditory Verbal Learning Test were not available for the HCs. The statistical comparisons are reported in the main text.

Variables	AD	MCI	HC	*F*_2,57_	*p*	AD vs. MCI	AD vs. HC	MCI vs. HC
**Gender (Males/Females)**	7/13	8/12	12/8					
	**Mean (SE)**	**Mean (SE)**	**Mean (SE)**					
**Age (years)**	72 (1.92)	72.20 (1.79)	70.35 (1.4)	0.35	0.71	-	-	-
**Education (years)**	9.65 (1.17)	11.95 (0.92)	11.60 (1.12)	1.32	0.27	-	-	-
**MMSE**	16.40 (1.05)	25.90 (0.43)	28.75 (0.29)	**91.26**	**<0.0001**	**<0.0001**	**<0.0001**	**<0.0001**
**HDRS**	10.5 (1.24)	8.65 (1.02)	7.45 (1.09)	1.86	0.16	-	-	-
**STAI Y-1**	39.11 (2.19)	34.15 (1.74)	33.4 (1.41)	2.92	0.06	-	-	-
**STAI Y-2**	41.22 (2.02)	37.2 (2.40)	33.65 (1.92)	3.09	0.06	-	-	-
**PSQI**	5.10 (0.61)	5.35 (0.78)	6.05 (0.68)	0.50	0.61	-	-	-
**RAVLT Ir**	14.84 (1.78)	25.15 (1.72)	-	-	-	**0.0002**	-	-
**RAVLT Dr**	0.58 (0.3)	2.06 (0.38)	-	-	-	**0.004**	-	-

Abbreviations: MMSE: Mini Mental State Examination; HDRS: Hamilton Depression Rating Scale; STAI Y-1 and Y-2: State Trait Anxiety Index; PSQI: Pittsburgh Sleep Quality Index; RAVLTir: Ray’s Auditory Verbal Learning Test immediate recall; RAVLTdr: Ray’s Auditory Verbal Learning Test delayed recall.

**Table 2 brainsci-07-00051-t002:** Mean and standard errors of the polysomnographic variables of Alzheimer’s disease (AD) patients, amnesic mild cognitive impairment (MCI) patients and healthy controls (HCs). The results of the one-way ANOVAs (*F* and *p* values) were also reported, with post hoc unpaired *t*-test (*p* values) when ANOVAs were significant (*p* ≤ 0.05). Significant differences between groups are indicated in bold.

Variables	AD	MCI	HC	*F*_2,57_	*p*	AD *vs.* MCI	AD *vs.* HC	MCI *vs.* HC
	**Mean (SE)**	**Mean (SE)**	**Mean (SE)**					
**Stage 1 latency (min)**	51.02 (13.43)	31.02 (4.69)	21.45 (4.96)	3.01	0.06	-	-	-
**Stage 2 latency (min)**	42.88 (13.21)	31.97 (4.57)	14.15 (3.79)	3.01	0.06	-	-	-
**Stage 1 (%)**	11.59 (2.39)	8.54 (1.24)	6.71 (1.04)	2.19	0.12	-	-	-
**Stage 2 (%)**	74.90 (2.65)	74.33 (2.13)	76.76 (1.78)	0.33	0.72	-	-	-
**SWS (%)**	0.14 (0.07)	0.13 (0.06)	0.95 (0.37)	**4.43**	**0.01**	**0.87**	**0.04**	**0.04**
**REM (%)**	13.58 (2.54)	16.98 (1.98)	15.93 (1.28)	0.76	0.47	-	-	-
**WASO (min)**	86.38 (11.37)	105.78 (11.97)	83.86 (8.24)	1.26	0.29	-	-	-
**Awakenings (#)**	17.25 (3.05)	19.25 (1.90)	20.30 (1.81)	0.44	0.64	-	-	-
**Arousals (#)**	35.70 (7.41)	28.65 (5.28)	32.25 (6.09)	0.31	0.73	-	-	-
**TST (min)**	260.67 (18.74)	261.67 (14.45)	300.12 (14.88)	1.94	0.15	-	-	-
**TBT (min)**	389.07 (16.79)	418.13 (21.21)	397.18 (11.97)	0.77	0.47	-	-	-
**SEI % (TST/TBT)**	66.74 (3.66)	63.94 (3.71)	75.23 (2.68)	3.02	0.06	-	-	-

Abbreviations: SWS: slow-wave sleep; REM: rapid eye movement; WASO: wake after sleep onset; TST: total sleep time; TBT: total bed time; SEI: sleep efficiency index.

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
