# Peer review of "In Search of Sleep Biomarkers of Alzheimer’s Disease: K-Complexes Do Not Discriminate between Patients with Mild Cognitive Impairment and Healthy Controls"

_brainsci, 2017, doi:10.3390/brainsci7050051_

Round 1

Reviewer 1 Report

I really appreciated tha paper by Reda et al. I think that their results could have a significant impact on the research about sleep and neurodegeneration.

I have a few comments. Taking into account that mild cognitive impairment (MCI) can include lots of conditions not related to a neurodegenerative process, i suggest Author to add a limitation to their study. Moreover, it has been established recently a condition named "mild cognitive impairment due to Alzheimer's Disease" (Albert et al., 2011), which can better differentiate the processs underlying this disease. Finally, the lack of follow-up of MCI patient is a firther limit of this report. I suggest a further study to evaluate longitudinally MCI patients in order to evaluate the converters to Alzheimer's Disease and to test whether the reduction of K-complexes is present only in the advances stages of AD. We need further characterizations of MCI patients due to AD, and considering that sleep may change early in the AD process it could be important also evaluate the modification of K-complexes.

Author Response

I really appreciated the paper by Reda et al. I think that their results could have a significant impact on the research about sleep and neurodegeneration.

R: Thanks to the reviewer for his/her comment on the possible relevance of our study

I have a few comments. Taking into account that mild cognitive impairment (MCI) can include lots of conditions not related to a neurodegenerative process, i suggest Author to add a limitation to their study. Moreover, it has been established recently a condition named "mild cognitive impairment due to Alzheimer's Disease" (Albert et al., 2011), which can better differentiate the processs underlying this disease. Finally, the lack of follow-up of MCI patient is a firther limit of this report. I suggest a further study to evaluate longitudinally MCI patients in order to evaluate the converters to Alzheimer's Disease and to test whether the reduction of K-complexes is present only in the advances stages of AD. We need further characterizations of MCI patients due to AD, and considering that sleep may change early in the AD process it could be important also evaluate the modification of K-complexes.

R: We have mentioned that the MCI may contain some conditions not related to the neurodegeneration and the recent establishment of the condition of "mild cognitive impairment due to Alzheimer's Disease". We also wish to thank the reviewer for the suggestion of a longitudinal observation of these patients. In fact, this study is ongoing and we have already recorded some converted and not-converted MCI patients after at least one year. However, we have too few patients for this kind of analysis. The general impression is that the drop of KCs’ rate again discriminates the converted vs. not-converted patients.

Reviewer 2 Report

Very timely and important paper. It does however over-interpret the findings and has significant omissions that need to be addressed. More specifically:

-Line 24: MCI is not a "preclinical" stage of AD. It's associated with cognitive and functional impairment. Preclinical denotes the stage in a disease prior to the appearance of symptoms.

-Line  40: "sleep alterations" is not a standard term, please rewrite

-Line 42: cognitive impairment may be worsened by sleep related increases in amyloid deposition too, I'd recommend changing i.e. (id est) for e.g. (exempli gratia) or including some reference to this very relevant and novel literature and the complexity of mechanisms involved in cognitive decline and sleep disturbances.

-Line 45: see previous comment

-Line 80: the "fall of KCs" is not a standard term but rather a term published by the authors of this paper on a previous publication. Please mention this fact and rephrase accordingly.

-Line 95: multi-domain MCI includes memory and other domains, please specify this.

-Table 1: are the authors sure that the difference in MMSE between MCI and AD patients is statistically significant? The means don't look that much different and the sample sizes are small.

-Table 1: please add information about gender  and some measure of dementia stage severity like the CDR or GDS.

-Line 164 more information needs to be included about the single scorer (did he/she score the other studies in HC and AD), was he/she blinded to diagnosis (probably not if only MCI cases were included, etc.

-Line 143: pulse oximeter can be used to screen for OSA and has shown to be useful for moderate severe OSA, but is questionable whether it can detect mild OSA which this cohort will have very likely (the prevalence of OSA in this age range is between 30-80% most of which is mild). Please add the ODI index and compare it between HC, MCI and AD groups as it could be an important confounder. Please add the absence of an adequate measurement of OSA with AHI, AHI3, AHI3a or AHI4% as a limitation to the study in the Discussion section.

-Line 190: one would expect differences between the AD group and the HC at least in WASO and sleep efficiency. Maybe the AD group's sleep is not that impaired? This needs to be highlighted in the results as it would be an unexpected finding.

-Line 226: absence of evidence is not evidence of absence. This study does not exclude the possibility of considering the fall of KCs as a biomarker. It's definitely a possibility that is suggested by the data, but the authors should not over-interpret the findings. Also the authors seem to suggest through the paper that MCI is an intermediate state on the path to AD but they forget to mention that not only MCI patients are amyloid positive or decline to AD. There's a possibility that some of the MCI patients included in this study will not decline to AD and this would explain the findings. This should be discussed and included as a limitation.

-Line 237: "consolidated AD" not a standard term

-Line 266: the authors are alluding to the Braak and Braak stages but they should remember that neurofibrillary tangles initiate in the locus coeruleus so their statement about subcortical is not strictly correct. Please include a  reference and the correct wording.

-Line 280: once again a slight over-interpretation of the findings. The MMSE is a tool for screening of cognitive impairment. I'd suggest the use of conditional more often and less boastful language

Author Response

Very timely and important paper. It does however over-interpret the findings and has significant omissions that need to be addressed. More specifically:

R: We also thank this reviewer for his/her positive comment on our study

-Line 24: MCI is not a "preclinical" stage of AD. It's associated with cognitive and functional impairment. Preclinical denotes the stage in a disease prior to the appearance of symptoms.

R: According to the appropriate suggestion of the reviewer we have changed the term.

-Line  40: "sleep alterations" is not a standard term, please rewrite.

R: Also rewritten according to the requested change.

-Line 42: cognitive impairment may be worsened by sleep related increases in amyloid deposition too, I'd recommend changing i.e. (id est) for e.g. (exempli gratia) or including some reference to this very relevant and novel literature and the complexity of mechanisms involved in cognitive decline and sleep disturbances.

R: Done. We have shortly made a reference to studies linking (functions of) sleep to amyloid deposition.

-Line 45: see previous comment

R: According to the appropriate suggestion of the reviewer we have changed the term.

-Line 80: the "fall of KCs" is not a standard term but rather a term published by the authors of this paper on a previous publication. Please mention this fact and rephrase accordingly.

R: We have changed into “drastic decrease of KCs’ rate”

-Line 95: multi-domain MCI includes memory and other domains, please specify this.

R: We have now specified.

-Table 1: are the authors sure that the difference in MMSE between MCI and AD patients is statistically significant? The means don't look that much different and the sample sizes are small.

R: We checked the analysis and confirm that the difference is actually significant

-Table 1: please add information about gender and some measure of dementia stage severity like the CDR or GDS.

R: We added information on gender. With respect to some other measures of dementia stage severity, unfortunately we have not CDR or GDS. We only have some measures of the neuropsychological assessment for AD/MCI patients. The revised version of the Table 1 contains the results of immediate recall and delayed recall at the Rey’s Auditory Verbal Learning Test (RAVLT), and the corresponding statistical comparison in the main text.

-Line 164 more information needs to be included about the single scorer (did he/she score the other studies in HC and AD), was he/she blinded to diagnosis (probably not if only MCI cases were included, etc.

R: The scorer (F.R.) was the same of our previous study. She was blind at the time of the original scoring of the three groups. Then, the data of AD and HC were analyzed in terms of EEG topography of sleep (i.e., the main goal of our previous study), while the comparisons of the frequency of KCs was only considered in the current article. We have clearly specified this in the revised version. 

-Line 143: pulse oximeter can be used to screen for OSA and has shown to be useful for moderate severe OSA, but is questionable whether it can detect mild OSA which this cohort will have very likely (the prevalence of OSA in this age range is between 30-80% most of which is mild). Please add the ODI index and compare it between HC, MCI and AD groups as it could be an important confounder. Please add the absence of an adequate measurement of OSA with AHI, AHI3, AHI3a or AHI4% as a limitation to the study in the Discussion section.

R: The reviewer is right. However, we have to specify that the final enrolment in the study was based on the evaluation of regular sleep–wake cycle and on the absence of self-rated sleep disorders. The presence of other sleep disorders was objectively evaluated by nocturnal sleep recordings, as a clinical routine. In case of sleep disorder and/or respiratory diseases and obstructive sleep apnoea syndrome (OSAS), subjects were excluded by subsequent analyses. None of the selected subjects had more than 5 events with oxygen saturation <90% per hour of sleep. This has been now clearly explained in the revised version of our article.

-Line 190: one would expect differences between the AD group and the HC at least in WASO and sleep efficiency. Maybe the AD group's sleep is not that impaired? This needs to be highlighted in the results as it would be an unexpected finding.

R: The reviewer is right. Actually, the polysomnographic (PSG) differences between groups are smaller than the ones reported in other studies. Meanwhile we were completing the present study, our project was still ongoing. We have right now a preliminary information on a larger sample with respect to PSG measures. We can confirm the current finding. A possible explanation may rely on the strict exclusion criteria with respect to the presence of sleep disorders. We have shortly commented this point in our revised discussion.

-Line 226: absence of evidence is not evidence of absence. This study does not exclude the possibility of considering the fall of KCs as a biomarker. It's definitely a possibility that is suggested by the data, but the authors should not over-interpret the findings. Also the authors seem to suggest through the paper that MCI is an intermediate state on the path to AD but they forget to mention that not only MCI patients are amyloid positive or decline to AD. There's a possibility that some of the MCI patients included in this study will not decline to AD and this would explain the findings. This should be discussed and included as a limitation.

R: This limitation has been included in the limitations to the study, also according to a similar comment of the other reviewer.

-Line 237: "consolidated AD" not a standard term

R: It has been changed

-Line 266: the authors are alluding to the Braak and Braak stages but they should remember that neurofibrillary tangles initiate in the locus coeruleus so their statement about subcortical is not strictly correct. Please include a  reference and the correct wording.

R: Thanks to the reviewer. We have corrected the wording and provided an appropriate reference

-Line 280: once again a slight over-interpretation of the findings. The MMSE is a tool for screening of cognitive impairment. I'd suggest the use of conditional more often and less boastful language

R: We have changed some sentences with a more hypothetical language

Round 2

Reviewer 2 Report

Most of my concerns have been addressed except for the one that relates to the measurement of sleep apnea severity. The authors don't specify why they use a non standard measurement of ODI (events with oxygen saturation <90% per hour of sleep). An OSA patient with good baseline oxygen saturation (e.g. 98%) can spend all night having hypopneas with 4% desaturation that never reach the 90% cutoff. Please clarify. This information should be included in the Results section in addition to the Discussion. If apnea severity was not adequately measured it should be stated in the Methods and stated as a Limitation of the study.

Author Response

The reviewer is right and we have better explained in the revised methods and discussion.

Although none had more than 5 events with oxygen saturation <90% per hour of sleep, our participants could have had hypopneas unrevealed by the oximetry measurement.

Round 3

Reviewer 2 Report

All my concerns have been addressed now.